# Effects of Blasting Vibrations on an Arch in the Jiaohuayu Tunnel Described by Energy Response Spectrum Analysis

**Shijun Hou** [1], **Feng Xie** [2], **Shuaikang Tian** [1] **and Shufeng Liang** [1,*]

1 School of Mechanics and Civil Engineering, China University of Mining and Technology-Beijing, Beijing 100083, China
2 North Blasting Technology Co., Ltd., Beijing 100097, China
* Correspondence: 201638@cumtb.edu.cn

**Abstract:** The vibrations caused by tunnel blasting strongly affects the construction safety and progress of the tunnel itself. The arch vibration attenuation law, structural energy response, and safety criterion were systematically investigated using blasting vibration monitoring in the Jiaohuayu Tunnel. The peak particle velocity (PPV) at the vault was always larger than that at the arch waist and was greater than that at the sidewall, regardless of the direction. The arch waist was where the initial lining had the highest risk of damage. Existing safety criteria can be supplemented and improved using the maximum instantaneous input energy to measure the first passage damage, the hysteretic energy consumption to measure the cumulative damage, and the input-hysteretic energy criterion to judge the structural failure. The energy threshold of the first passage damage of the initial lining structure was 200 J, and the plastic cumulative damage was 3000 J of the test section. This study is important when evaluating the safety of a tunnel's initial lining structure.

**Keywords:** blast vibration; tunnel arch; energy response spectrum; maximum instantaneous input energy; hysteretic energy consumption

## 1. Introduction

With a large-scale construction project that is part of China's transportation infrastructure, the corresponding tunnel construction projects are increasing. Drilling and blasting are still the primary methods of tunnel excavation due to their economic and efficiency characteristics. However, blasting vibrations may damage the tunnel's support structure and cause a collapse of the adjacent surrounding rock.

The blasting seismic wave will attenuate as it propagates [1,2]. Due to variable and uncertain geological conditions in the tunnel, this attenuation is complex. Many scholars have studied vibration attenuation inside tunnels to improve structural safety and stability [3–5]. Clarifying which segment blasting has the strongest impact on vibration and what the influencing factors are important to tunnel blasting design, real vibration reduction, and disaster prevention. More importantly, the response mechanism of different engineering structures to blasting vibration is different. The blast centre conditions and the properties of the measured structure will affect the dynamic response characteristics of the structure. Heelan [6] and White [7] used the superposition method and reciprocity theorem to calculate the response of columnar and slender chambers under finite long-term dynamic loads, respectively. Kong [8] studied the impact response of subway tunnels under blast loading by numerical simulation, which provided a reference for blast-resistant protection and emergency plan for subway tunnels. Li et al. [9] studied the effect of tunnel blasting vibration on the excavated part of the lining of different ages through model tests. Xiong et al. [10] analysed the damage characteristics of tunnel lining structures based on numerical simulation and the energy principle. Lu et al. [11] found that the response spectrum analysis method can be used to investigate the dynamic response of tunnel



structures under blasting vibration. Various structures respond differently to blast seismic waves; thus, it is necessary to quantify the effects of the inherent properties of the vibrated structure's response to blasting vibration.

Chinese safety regulations for blasting vibration [12] stipulate that the PPV and the dominant frequency are used as the comprehensive criterion for the tunnel structure. However, the tunnel structure is sometimes damaged when the PPV is much lower than the safe allowable PPV, which indicates that the current safety criteria are imperfect. Lou et al. [13] proposed that the integral value of the response spectrum curve be used as the evaluation standard for the damage effect of blasting vibration. Zhang [14] proposed the equivalent velocity under the maximum instantaneous input energy as the blasting vibration safety criterion based on the HHT (Hilbert–Huang Transform) method. Li [15] proposed the evaluation index of EPE (Equivalent Peak Energy) and preliminarily discussed the possibility of establishing the EPE evaluation index. Ling [16] established the integral value of the TEDI (Integral of the Time–Energy Density) based on the wavelet time–energy density method as a unified safety criterion for blasting vibration damage and determined the damage threshold of TEDI > 15 through a neural network. That research provides a basis for blasting vibration safety criterion and highlights the direction for follow-up safety criterion research. However, the energy response of the tunnel structure under the action of blasting vibration is affected by the blasting seismic wave characteristics and the inherent properties of the structure itself.

Based on the above analysis, the safety criterion based solely on the characteristics of blasting seismic waves or the inherent properties of the structure markedly cannot fully describe the real hazards of engineering structures under the action of blasting seismic waves. Therefore, this paper takes the Jiaohuayu tunnel of the Xingyan highway as the engineering background and uses blasting vibration monitoring to study the blasting vibration effect in the tunnel, which can make up for the deficiency of the previous evaluation method of the blasting vibration hazard effect.

## 2. Engineering Background

The Xingyan highway is a key supporting project for the 2022 Beijing Winter Olympics that connects Beijing City and Yanqing District with a total length of approximately 42.2 kilometres. The test site is located in the shallow buried section from YK28 + 190 to YK28 + 354 of the Jiaohuayu right line tunnel. The Jiaohuayu right line tunnel is located in the alternation area of the middle-low mountain landform unit and the intermountain valley landform unit. The surrounding rock within the scope of the test site is mainly slightly to moderately weathered dolomite with grade level III, locally distributed neutral intrusive dykes according to the results of geological exploration. Rock joints and fissures are generally developed.

The two-step blasting method was used for construction, in which the upper step used smooth blasting and the lower step used horizontal hole grooving blasting. Because the excavation surfaces of the upper and lower steps were more than 70 m apart, the upper step blasting had the characteristics of full-section blasting. The excavation area of the tunnel face was 90.5 m$^2$, while the lower step was 59.5 m$^2$. The field test primarily monitored the vibration of the upper blasting step. The blast holes of each part of the tunnel blasting were divided more carefully to meet the requirements of the blasting construction of the large section tunnel. Figure 1 shows the layout of the arrangement and timing sequence of the upper step blasthole. Cut holes were first detonated so as to create a new free surface for other blasting holes and reduce the clamping of other blasting holes. Auxiliary hole and caving hole blasting were followed successively, playing a role in expanding the cutting. Finally, contour blasting was used to make the tunnel section, shape, and direction meet the design requirements after blasting. Not only the characteristics of the tunnel project and the surrounding environment were considered, but also the construction experience and design formula of similar projects were used for reference in the design of the Jiaohuayu tunnel blasting scheme.

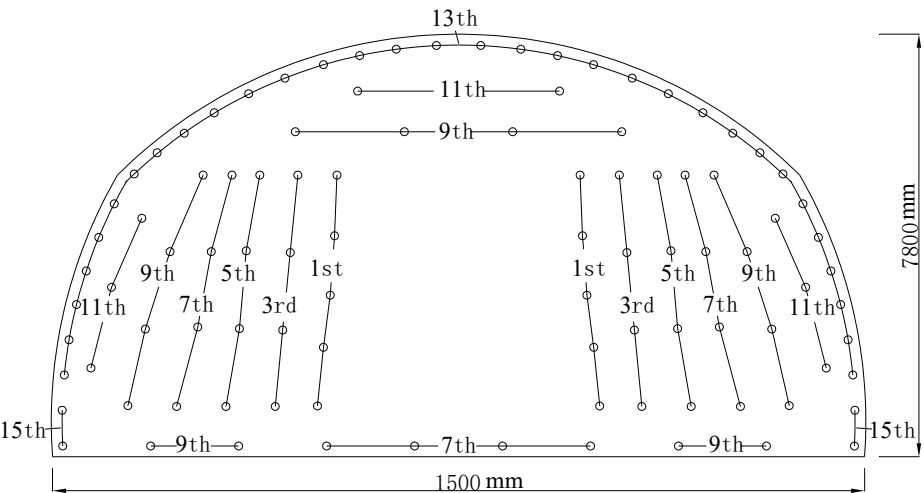

**Figure 1.** Blasthole arrangement and detonation sequence diagram.

## 3. Vibration in the Tunnel Arch

### 3.1. Blasting Vibration Monitoring Scheme

To comprehensively describe the vibrations in the arch structure during tunnel drilling and blasting construction, three measurement points are arranged in the vault, arch waist, and sidewall 38 m away from the tunnel face. The layout diagram of the tunnel vibration measurement points is shown in Figure 2.

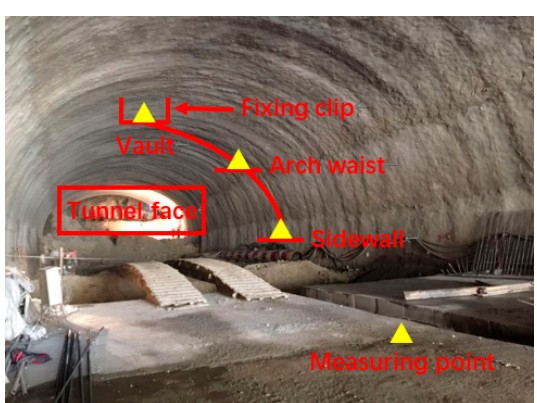
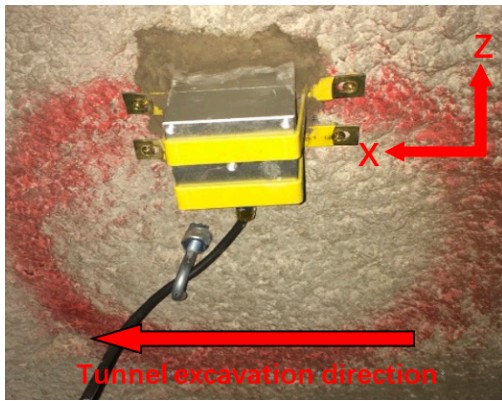

**Figure 2.** Measuring point arrangement diagram on the arch structure in the tunnel.

TC-4850 blasting vibrometers were used to collect the blasting vibration data. The sampling frequency of the vibrometer is 1~50 kHz, and the frequency response range is 5~500 Hz. The recording accuracy is 0.01 cm/s. TCS-B3 low-frequency broadband three-dimensional vibration velocity sensors were used to monitor the blasting vibration. When arranging measuring points in the tunnel arch, a cement anchoring agent was used due to its characteristics of rapid setting, high bonding strength, and good durability. The cement anchoring agent can be used after soaking in water for 30–40 s. The initial lining structure surface of the tunnel at the measuring point was smoothed with a cement anchoring agent; then, the TCS-B3 three-dimensional vibration velocity sensor was placed on it with a fixing clip so as to ensure that the sensor and the initial lining structure share the overall vibration.

### 3.2. Attenuation Law of the Blasting Vibration Velocity

To describe the attenuation law of the blasting vibration velocity, the Sadowski formula is a common method. The Sadowski formula is:

$$V = K \cdot (R/Q^{1/3})^{\alpha} \tag{1}$$

where $V$ is the PPV (cm/s); $Q$ is the number of explosives (kg); $R$ is the distance from the blast centre (m); and $K$ and $\alpha$ are the coefficient and attenuation index related to terrain and geological conditions, respectively. Reference [17] shows that the Sadowski formula determined by the nonlinear regression method has higher accuracy, and then, $K$ and $\alpha$ can be obtained. The PPVs of the vertical, horizontal tangential and horizontal radial directions with proportional distances at the tunnel vault, arch waist, and sidewall are fitted.

Figure 3 and Table 1 show that the PPV at the vault was always larger than at the arch waist, which was greater than at the sidewall regardless of the vertical, horizontal tangential, or horizontal radial direction. With an increasing proportional distance, the PPV gradually approaches, which indicates that the three-way vibration velocity may overlap or even change the size relationship with a further increase in the proportional distance. The vertical, horizontal radial, and horizontal tangential $K$ of the vault vibration velocity are all larger than the corresponding components of other test positions. The vertical vibration velocity is the largest. Although the shock wave is attenuated by the propagation path, the residual blasting energy of the vault is higher than that of the other test positions. The $\alpha$ of the three components of the vibration velocity of the measuring points at the same position is close. In addition, the absolute values of the vertical, horizontal radial, and horizontal tangential $\alpha$ of the vault are larger than the corresponding components of other blasting holes, indicating that the vibration velocity of the vault decays faster.

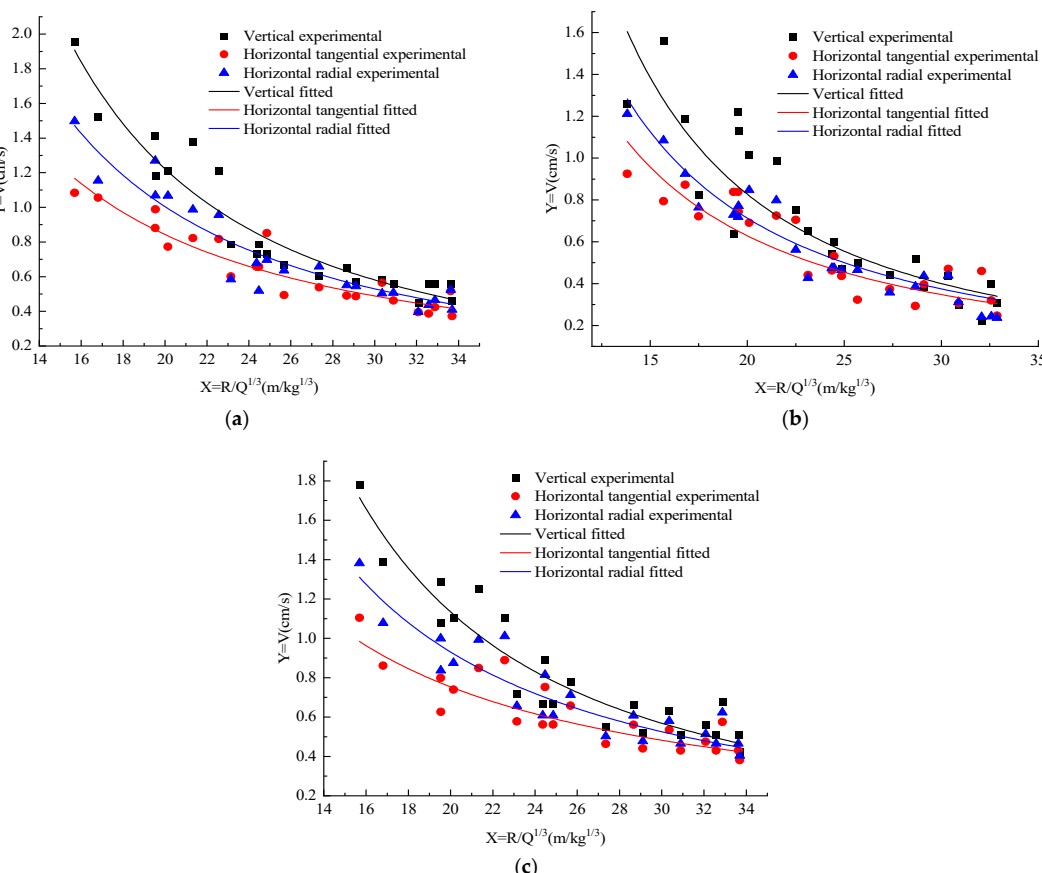

**Figure 3.** Three-component vibration velocity attenuation law and regression curve. (**a**) At vault; (**b**) At arch waist; (**c**) At sidewall.

**Table 1.** Comparison of *K* and *α* in three directions.

| Measuring Point Location | K | | | α | | |
|---|---|---|---|---|---|---|
| | Vertical | Horizontal Tangential | Horizontal Radial | Vertical | Horizontal Tangential | Horizontal Radial |
| Vault | 296.2 | 126.7 | 204.3 | 1.95 | 1.91 | 1.92 |
| Arch waist | 235.1 | 123.2 | 194.2 | 1.86 | 1.83 | 1.85 |
| Sidewall | 183.7 | 96.4 | 108.3 | 1.71 | 1.69 | 1.70 |

### 3.3. Dominant Frequency Analysis of Vibration

Fifty groups of vertical dominant frequencies were selected for statistical analysis to better understand the characteristics of tunnel blasting vibration. According to the frequency prediction relationship deduced by the dimensional theory in [18], the regression statistics of the vibration frequency of the tunnel blasting are performed, which is shown in Figure 4. The correlation coefficients of the dominant frequency regression curves at each position are all near 0.6. There are differences in the dominant frequency at different positions of the tunnel arch, and the size relationship is vault > sidewall > arch waist. In addition, in the process of tunnel blasting, millisecond delay detonator blasting vibration signal energy was primarily concentrated within 200 Hz. Based on the distribution law of the vibration velocity of the arch, the vibration velocity of each position is not much different as the distance between the measuring point and the blasting centre increases. When the vibration velocity is similar, the dominant frequency of the arch waist position is the lowest. Therefore, the risk of damage at the arch waist position is highest if the damage occurs farther away from the tunnel face.

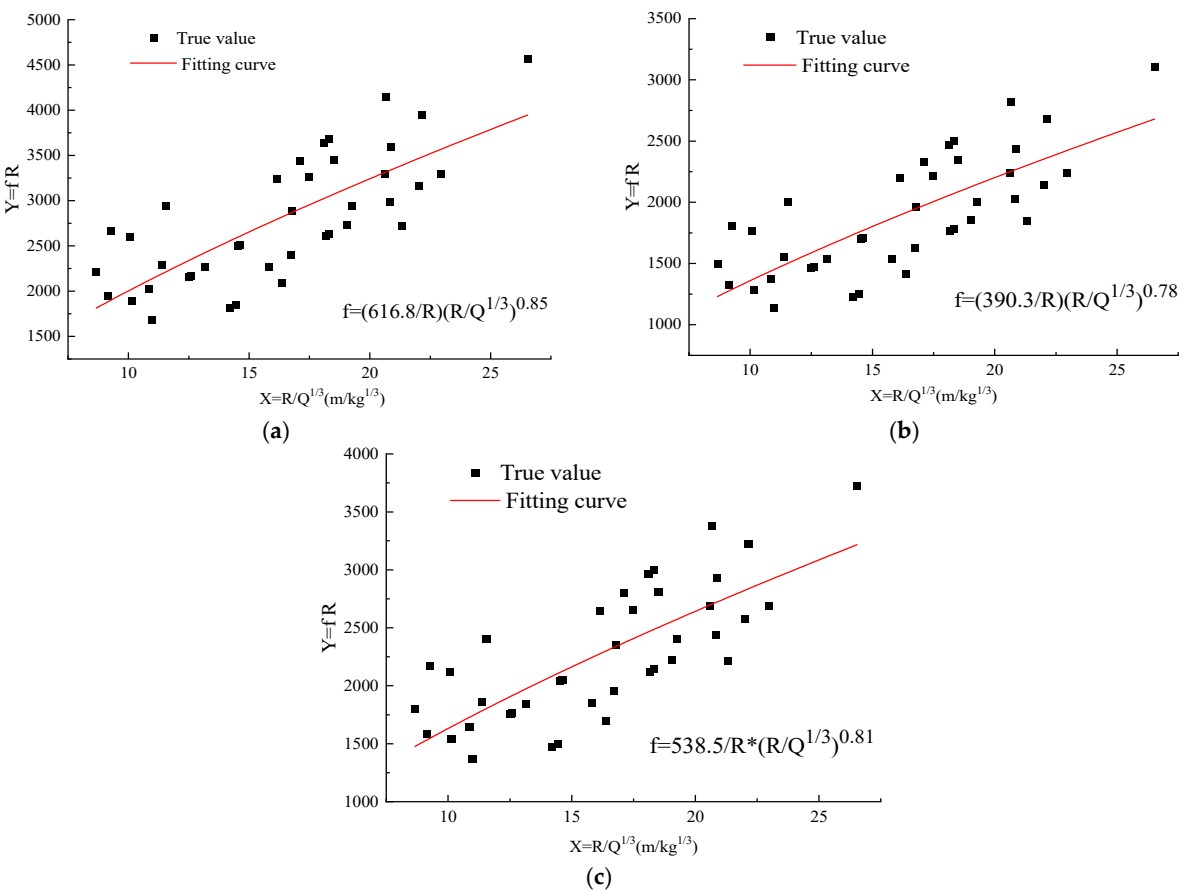

**Figure 4.** Statistical law of vertical dominant frequency of the tunnel at arch structure. (**a**) At vault; (**b**) At arch waist; (**c**) At sidewall.

## 4. Structural Energy Response Spectrum of the Tunnel Blasting Vibration

The influencing factors of the structural input energy include the PPV, dominant frequency, duration of blasting vibration, and the inherent parameters of the structure, such as the damping ratio, natural period, and hysteresis model parameters [19,20]. Therefore, the influence of three blasting vibration elements and structural inherent parameters on the input energy spectrum in an elastic SDOF system is studied. The measured blasting vibration velocity data should be first converted into acceleration signals, and then the energy spectrum should be calculated [21–25]. The formula to convert vibration data into an acceleration signal is:

$$a(t_i) = \frac{\Delta v(t_i)}{\Delta t_i} = \frac{v(t_{i+1}) - v(t_i)}{t_{i+1} - t_i} \tag{2}$$

However, the error order of Equation (2) is high with $o(\Delta t_i)$ when expanding it according to the Taylor formula, which will cause a large random error. The four-point forward difference method in [26], in which the error order is $o(\Delta t_i^3)$, can solve this problem effectively:

$$a(t_i) = \frac{2v(t_{i+3}) - 9v(t_{i+2}) + 18v(t_{i+1}) - 11v(t_i)}{6(t_{i+1} - t_i)} \tag{3}$$

Three waveforms with typical time–frequency characteristics are selected. The three acceleration signals obtained by calculation and EMD denoising are used as the reference signals to study the influence of PPV, dominant frequency, and duration on the energy spectrum. The acceleration signal and its Hilbert spectrum are shown in Figure 5. The dominant frequency of the first acceleration signal is 28.6 Hz. The energy of the acceleration signal is primarily distributed in 7~110 Hz, and the duration of the entire vibration is 1.18 s. The dominant frequency of the second acceleration signal is 37.1 Hz, and the energy is primarily distributed in the frequency range of 5~160 Hz, with a duration of 1.20 s. The dominant frequency component of the third acceleration signal is more abundant than that of the first two. The energy is primarily distributed in 5~180 Hz with a dominant frequency of 50 Hz, and its duration is 1.16 s. The three reference signals are adjusted manually, and the influence of one factor on the energy spectrum is studied using the control variable method and fixing all other factors.

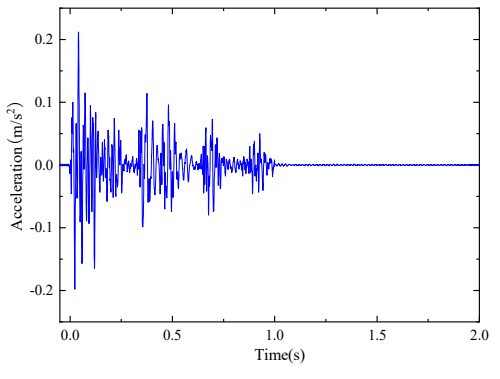 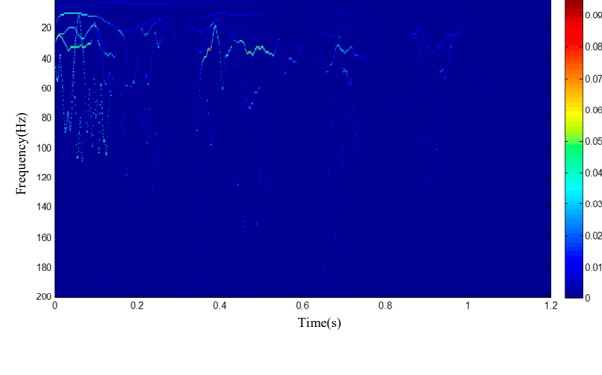

(a)

**Figure 5.** *Cont.*

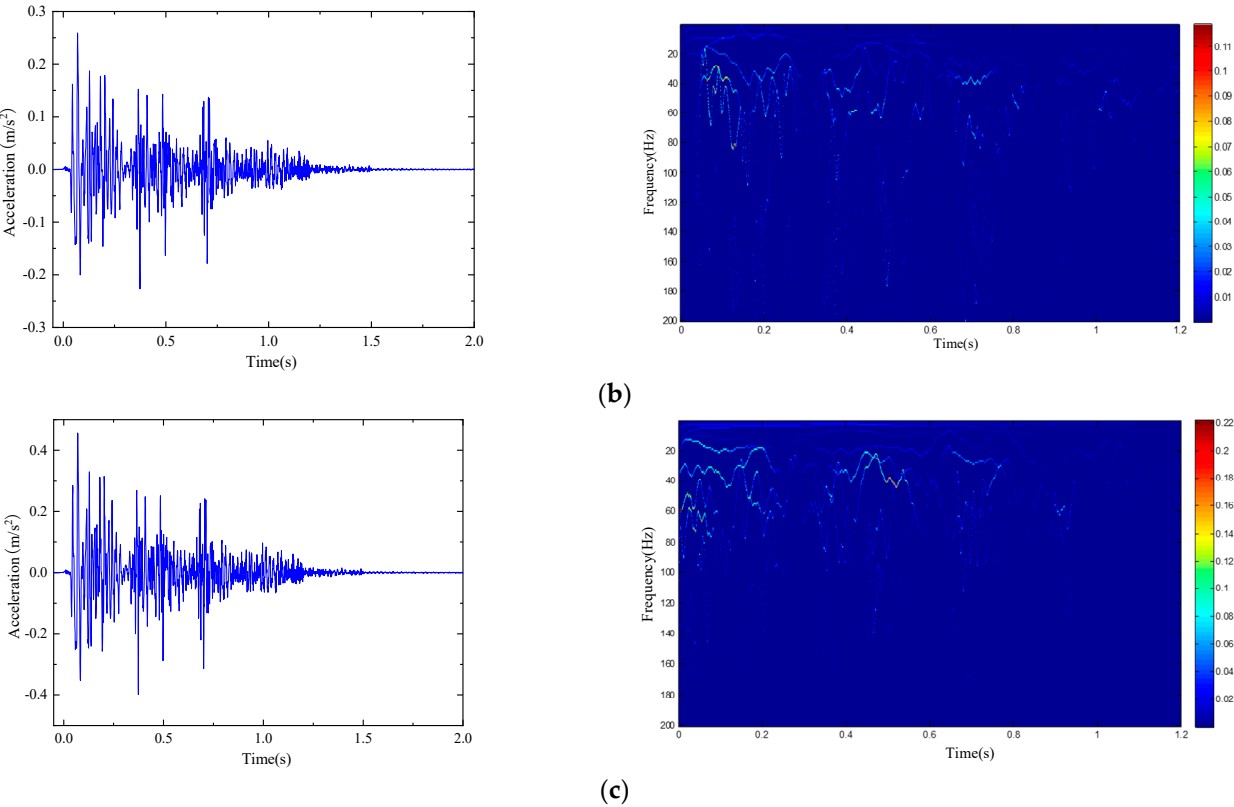

**Figure 5.** Acceleration signals and its Hilbert spectra. (**a**) With a dominant frequency of 28.6 Hz; (**b**) With a dominant frequency of 37.1 Hz; (**c**) With a dominant frequency of 50.0 Hz.

### 4.1. Influence of Blasting Vibration Characteristics on the Input Energy Spectrum

The acceleration signal in Figure 5a is selected as the input data of the structural dynamic calculation. The natural period and damping ratio of the structure are selected as 0.1 s and 0.05, respectively. The acceleration, velocity, and displacement time history curves of the tunnel structure under blasting vibration, as shown in Figures 6–8, are obtained by the Newmark-β step-by-step method. Then, the time history curves of input energy $E_I$, kinetic energy $E_K$, elastic strain energy $E_E$, damping energy $E_D$ and the ratio of different forms of energy to total energy are calculated by the method in [27], as shown in Figures 9 and 10, respectively. The natural period of the structure is defined within 0~1.0 s, and the time interval is selected as 0.01 s. The input energy of the structure under different periods is calculated, which is shown in Figure 11.

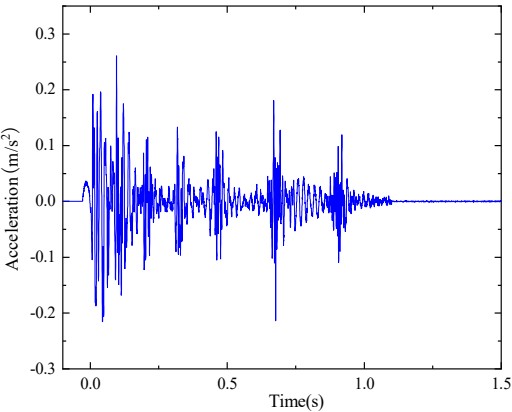

**Figure 6.** Time history curve of the elastic response acceleration.

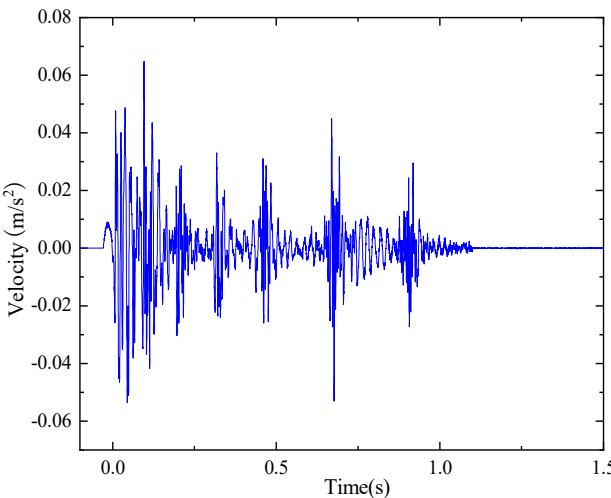

**Figure 7.** Time history curve of the elastic response velocity.

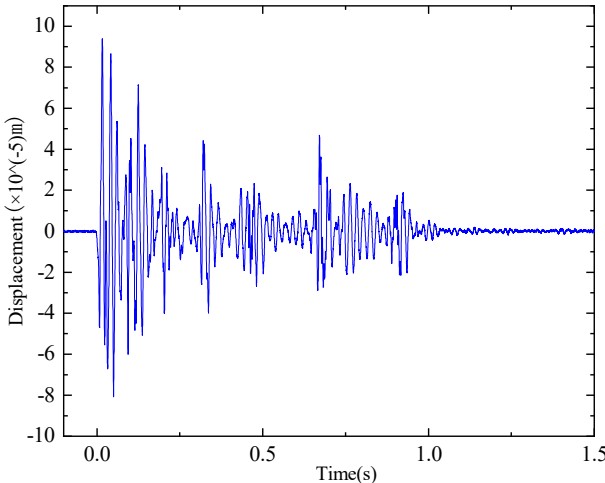

**Figure 8.** Time history curve of the elastic response displacement.

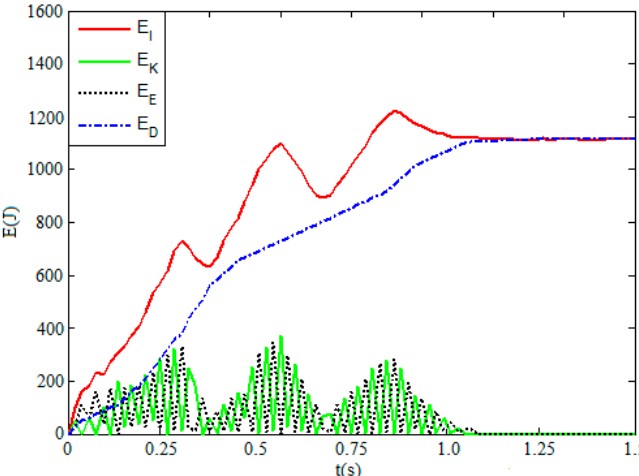

**Figure 9.** Time history curve of the elastic response energy.

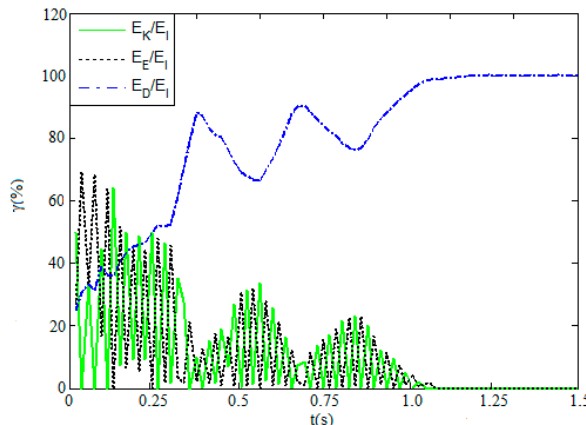

**Figure 10.** Ratio of each energy to the input energy.

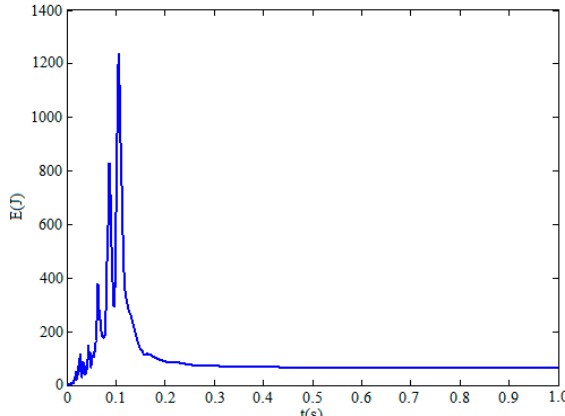

**Figure 11.** Input energy spectrum.

Figures 9 and 10 show that the input energy basically increases with the vibration time, and the input energy generally reaches the peak when the vibration ends. However, parts of the input energy curve are declining, which is partly due to the hysteresis of the structural dynamic response caused by damping, and also due to the integral definition of the input energy causing the blasting vibration force to be negative. The kinetic energy and elastic strain energy will increase with time in the elastic SDOF system and also transform each other in the process. Finally, all the energy is converted into damping energy and dissipated; thus, the damping energy dissipation reflects the cumulative effect over time.

To compare the influence of the acceleration peak on the input energy spectrum, the peak acceleration signal in Figure 5a is adjusted to $0.2$ m/s$^2$, $0.5$ m/s$^2$, and $0.8$ m/s$^2$, and its input energy spectrum with a damping ratio of 0.05 is shown in Figure 12. Results show that the change in the acceleration peak does not affect the shape of the input energy spectrum. The peak input energy increases with increasing peak acceleration in the square.

The peak acceleration signal is normalised to analyse the influence of the dominant frequency on the input energy spectrum. The input energy spectrum of the acceleration signal at three dominant frequencies with a damping ratio of 0.05 is shown in Figure 13, which shows that the input energy spectrum varies markedly with the dominant frequency of the signal. The peak value of the input energy spectrum and its corresponding structural natural period decrease as the dominant frequency increases. The number of peak input energies with a dominant frequency of 50 Hz is more than that of other waves. The larger the dominant frequency is, the richer the frequency component of the vibration waveform is. The dominant frequency of other blasting waveforms is near that of the full-time waveform; thus, there are many peaks.

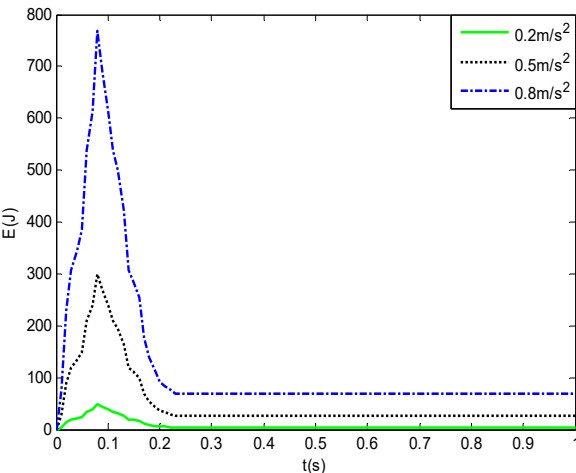

**Figure 12.** Influence of the acceleration peak on the input energy spectrum.

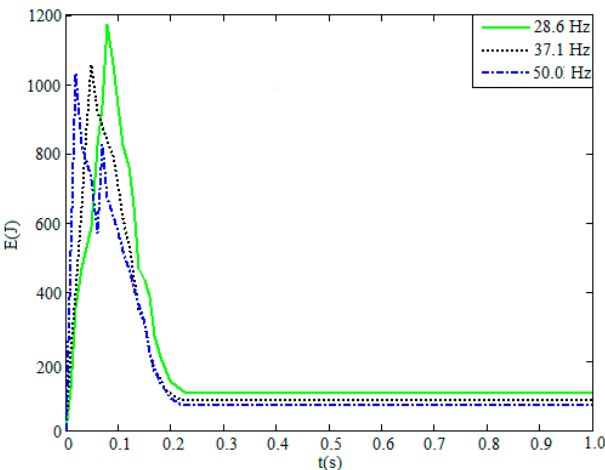

**Figure 13.** Influence of the dominant frequency on the input energy spectrum.

The acceleration signal in Figure 5a is selected to analyse the influence of the blasting vibration duration on the input energy spectrum. The duration is adjusted by doubling or tripling the acceleration signal, and the peak acceleration of each segment signal is normalised. The damping ratio is 0.05. Figure 14 shows that the change in duration has no effect on the shape of the input energy spectrum, but the smoothness of the curve is markedly deteriorated. Because the external vibration has been inputting energy for the structure, the peak input energy increases with increasing duration, and this increasing trend ends when the vibration stops. The input energy spectrum can accurately describe the vibration factor of duration and can accurately integrate the response law of the PPV, dominant frequency, and duration of vibration to the structure, which is an important reference that should be used to improve safety criteria.

Excluding the influence of PPV, dominant frequency, duration of blasting vibration waves, and other structural factors, the influence of the damping ratio on the energy spectrum can be studied. Figure 15 shows the input energy spectra with damping ratios of 0.02, 0.05, and 0.08. The larger the damping ratio is, the smaller the peak value of the corresponding input energy spectrum is. This result is primarily due to a large part of the energy consumed by damping and the ratio of the total energy occupied by the kinetic energy and strain energy that can be transformed into each other decreases. Therefore, the peak input energy decreases, but in the area outside the curve peak, the change in the input energy is the opposite: the larger the damping ratio, the smoother the input energy spectrum curve, which is consistent with the research results in [28,29].

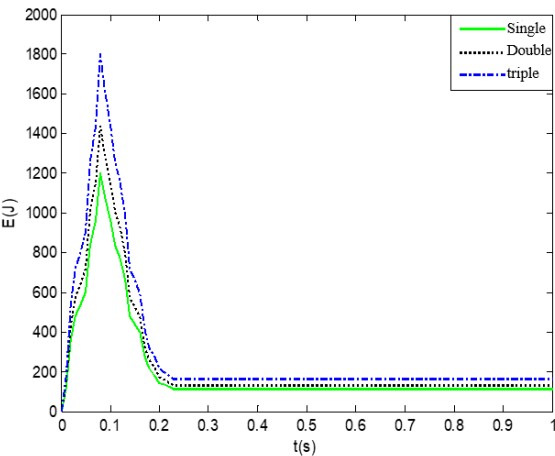

**Figure 14.** Influence of the duration on the input energy spectrum.

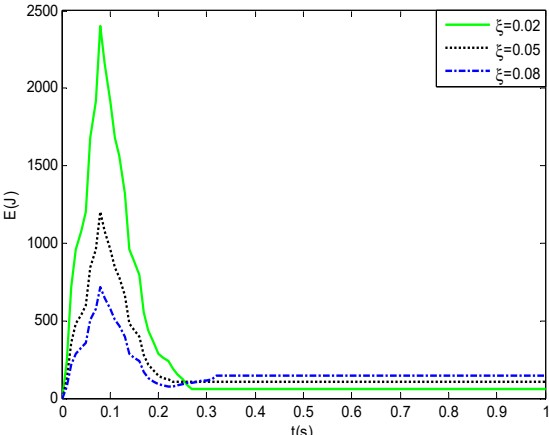

**Figure 15.** Influence of the damping ratio on the input energy spectrum.

The input energy time history curve is shown in Figure 16 with natural periods of 0.02 s, 0.04 s, and 0.08 s to analyse the influence of the natural period of the structure on the input energy. The natural period of the structure has no effect on the shape of the input energy spectrum. The period corresponding to the dominant frequency of the original waveform is 0.035 s. When the characteristic period of the blasting seismic wave is consistent with or near the natural period of the structure, it is closest to the resonance. Therefore, the maximum input energy at 0.04 s is the largest of the three input energy spectra.

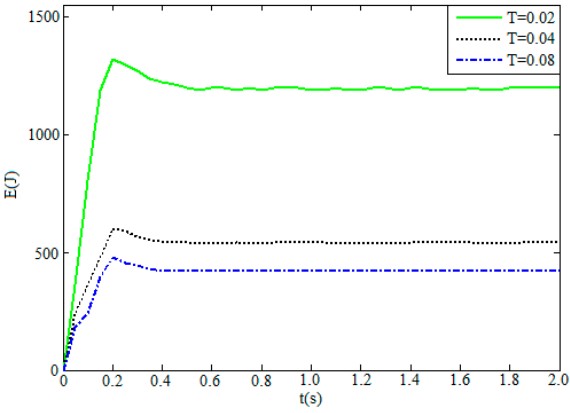

**Figure 16.** Influence of the natural period on the input energy spectrum.

*4.2. Influence of Blasting Vibration Characteristics on the Hysteretic Energy Consumption Spectrum*

The biggest defect of the energy response spectrum of the elastic SDOF system structure is that it cannot reflect the cumulative effect of blasting vibration. In tunnel blasting, the cumulative effect is primarily manifested in multiple repeated blasting and blasting with a long duration. Because the tunnel face must be continuously drilled, the same tunnel structure must be affected by multiple blasting vibrations. Multiple detonators are usually used in tunnel blasting, and the duration will be longer. Because the damping ratio typically appears in the form of a fixed value in the calculation process, the energy consumed by plastic deformation is not involved in the conversion of energy and is finally dissipated completely. Therefore, when evaluating the cumulative damage of engineering structures [30,31], it is of practical importance to measure the cumulative damage of structures with hysteretic energy consumption.

Excluding the influence of the dominant frequency, duration, structural parameters, and restoring force model parameters, the influence of the PPV on the hysteretic energy spectrum is analysed. Using the three acceleration time history curves in Figure 5, the hysteretic energy spectrum is shown in Figure 17 with a damping ratio of 0.05, a yield strength coefficient of the bilinear restoring force model of 0.3, and a stiffness reduction coefficient after yielding of 0.02. Figure 17 shows that the increase in the peak acceleration has no effect on the shape of the input energy spectrum. The square of the increase of acceleration is the increase of hysteretic energy consumption of the system, which indicates that the increase of acceleration will lead to the rapid increase of the input energy of the system.

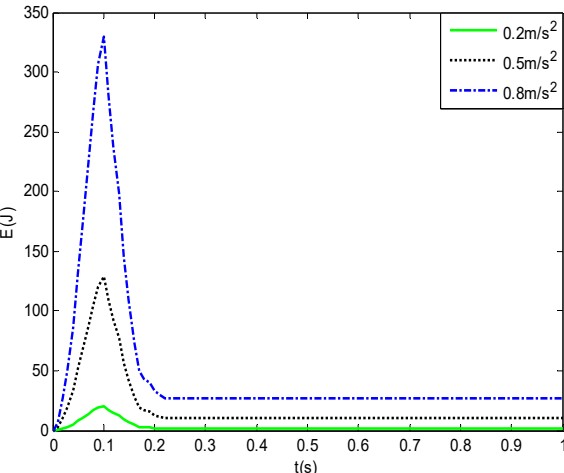

**Figure 17.** Influence of the acceleration peak on the hysteretic energy consumption.

The same method is used to investigate the influence of the dominant frequency on the hysteretic energy spectrum. Figure 18 shows that the natural period of the structure corresponding to the peak of hysteretic energy dissipation decreases with the increasing dominant frequency of the blasting seismic wave. There are multiple peaks on the hysteresis energy spectrum of the blasting seismic wave with a dominant frequency of 50 Hz, which is the same as the influence of the dominant frequency on the input energy spectrum. There is a close relationship between the dominant frequency of the tunnel-blasting seismic wave and the shape of the hysteretic energy dissipation spectrum.

Figure 19 shows the influence curve of duration on the hysteretic energy spectrum. The peak hysteretic energy increases as the duration increases, which shows that the hysteretic energy consumption is a process of time accumulation. The longer the duration of the vibration is, the greater the value is. Therefore, hysteretic energy consumption can accurately reflect the cumulative effect of duration on structural plastic failure. The vibration factor of duration is included in the scope of plastic cumulative damage of the

structure; thus, the hysteretic energy dissipation as the criterion of cumulative damage can fully describe the vibration's characteristics.

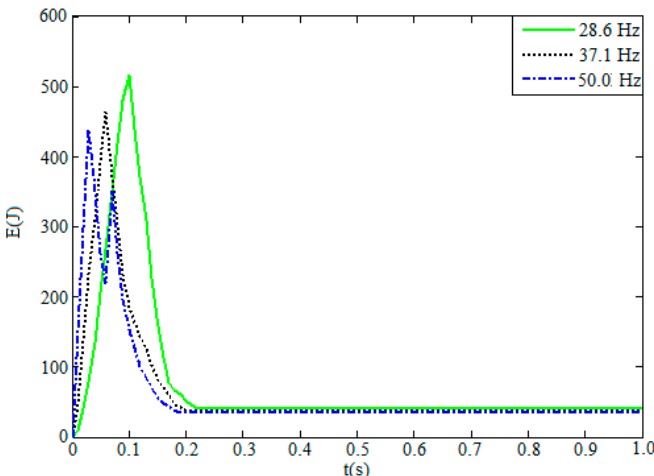

**Figure 18.** Influence of the dominant frequency on the hysteretic energy consumption.

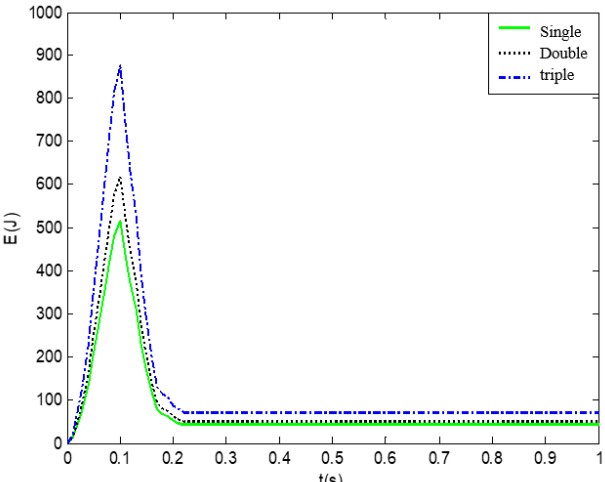

**Figure 19.** Influence of the duration on the hysteretic energy consumption.

*4.3. Influence of Structural Parameters and Restoring Force Model Parameters on the Hysteretic Energy Consumption Spectrum*

To analyse the influence of the damping ratio on the hysteretic energy spectrum, the influence of the blasting vibration characteristics and restoring force model parameters should be excluded. The acceleration signal in Figure 5a is selected, and the peak value is normalised. The yield strength coefficient of the bilinear restoring force model is 0.3. The stiffness reduction coefficient after yielding is 0.02. The structural damping ratio is set as 0.02, 0.05, and 0.08. Figure 20 shows that the shape of the hysteretic energy spectrum curve does not change with increasing structural damping ratio. The peak energy of the hysteretic energy spectrum increases when the damping ratio decreases. Thus, when the input energy is constant, the greater the damping ratio is, the greater the damping energy is, and the corresponding hysteretic energy will decrease. Evidently, setting obstacles on the propagation path of the blasting seismic wave can reduce the probability of cumulative damage or failure of the engineering structure.

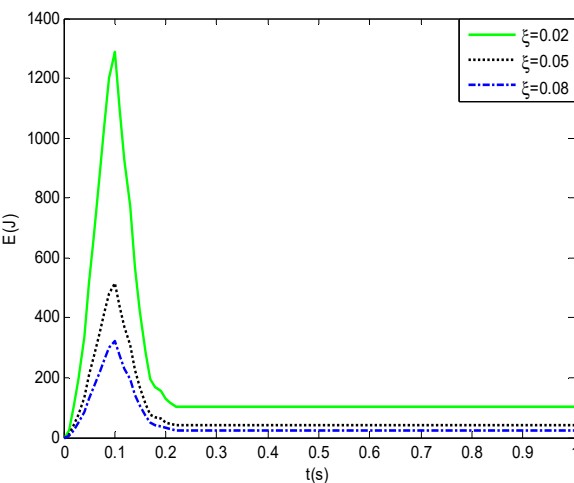

**Figure 20.** Influence of the damping ratio on the hysteretic energy consumption.

The influence of the natural period of the structure on the hysteretic energy dissipation is analysed using the same method. The natural periods are set as 0.02 s, 0.04 s, and 0.08 s, respectively. Figure 21 shows that the better the coupling between the vibration predominant period of the blasting seismic wave and the natural period of the structure, the greater the hysteretic energy consumption. Considering the resonance when the two are completely coupled, this is the case where the hysteretic energy consumption will have an extreme value, that is, the greatest threat to the related engineering structure.

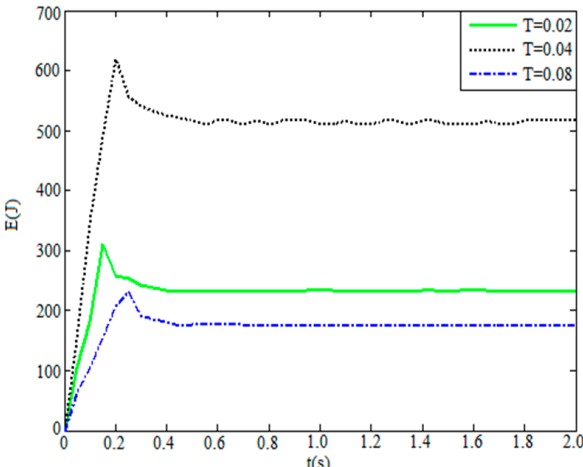

**Figure 21.** Influence of the natural period on the hysteretic energy consumption.

The influence of the yield strength coefficient and stiffness reduction coefficient on hysteretic energy consumption is analysed by the same method. The stiffness reduction factor of the bilinear restoring force model after yielding is 0.02, the yield strength coefficients are 0.15, 0.25, and 0.3, and the hysteretic energy spectrum is shown in Figure 22. The yield strength coefficient of the bilinear restoring force model is 0.3, the stiffness reduction coefficients are 0, 0.02, and 0.05, and the hysteretic energy spectrum is shown in Figure 23. Figures 22 and 23 show that the increase or decrease in the yield strength coefficient and the stiffness reduction coefficient does not affect the shape of the hysteretic energy consumption spectrum, nor does it cause a drastic change in the hysteretic energy consumption value. The two curves nearly coincide, and the smoothness of the curve does not change. Therefore, the parameters of the restoring force model are not the influencing factors that change the shape of the hysteretic energy dissipation spectrum.

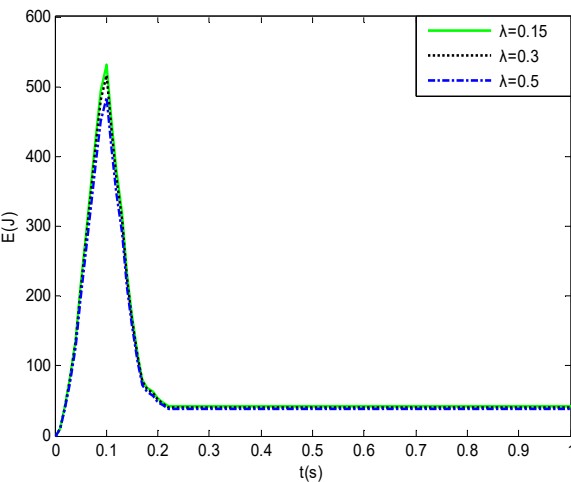

**Figure 22.** Influence of the yield strength coefficient on the hysteretic energy consumption.

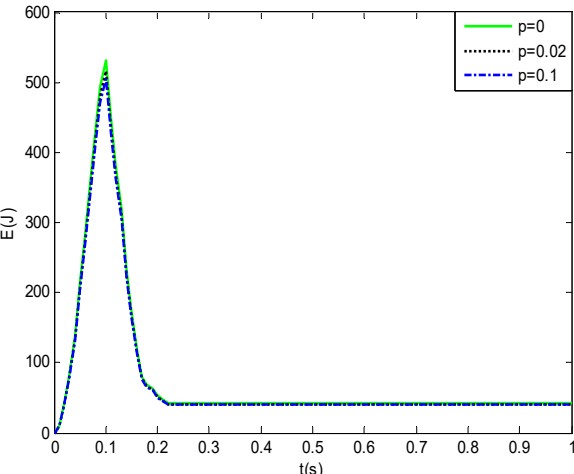

**Figure 23.** Influence of the stiffness reduction coefficient on the hysteretic energy consumption.

*4.4. Influence of Blasting Seismic Waves on the Hysteretic Energy Consumption Spectrum under Different Blasting Conditions*

Excluding the influence of distances from the blasting centre, delay time, and site factors, the influence of charge weight on the hysteretic energy spectrum is analysed. The charge weights of the cut hole are 38 kg, 30 kg, 24 kg, and 20 kg. The damping ratio of the structure is 0.05. The yield strength coefficient of the bilinear restoring force model is 0.3, and the stiffness reduction coefficient after yielding is 0.02. Figure 24 shows that the peak energy of the hysteretic energy spectrum increases markedly with increasing segment charge. The segment charge is an important factor that affects the hysteretic energy consumption and provides theoretical support for the generally accepted measure of controlling the segment charge to achieve safe production in real blasting.

The influence of the distance from the blasting centre on the hysteretic energy spectrum is analysed in the same way. The charge weight of the cut hole is 30 kg, and the distances from the blasting centre are 38 m, 48 m, 68 m, and 98 m, respectively. Figure 25 shows that the change in distance from the blasting centre has no effect on the shape of hysteretic energy consumption, which is reflected in the smoothness of the hysteretic energy spectrum curve. It can be explained that the distance from the blasting centre is not the influencing factor of the shape change of the hysteretic energy spectrum. The increase in the distance from the blasting centre will lead to a decrease in the peak hysteretic energy dissipation, PPV, and dominant frequency, while the duration of the vibration will be prolonged. The

combined effect of these factors leads to the peak hysteretic energy not decreasing markedly, which may lead to the structure near the explosion source not being damaged or destroyed while the structure far from the blasting centre is being destroyed.

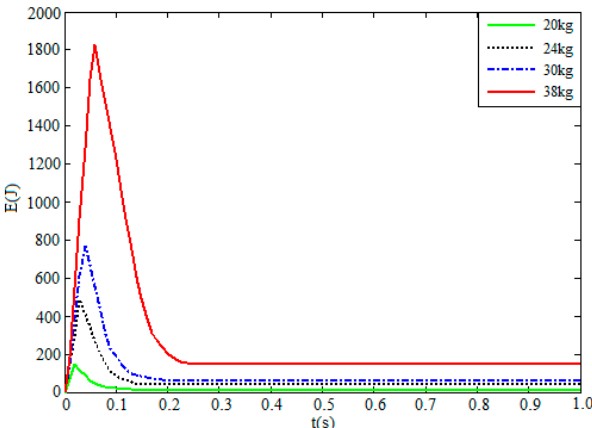

**Figure 24.** Influence of the segment charge on the hysteretic energy consumption.

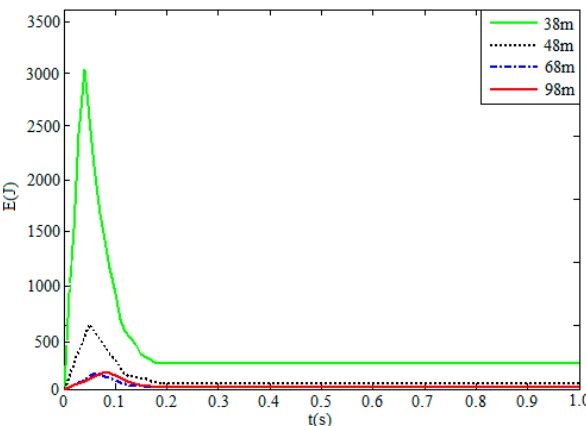

**Figure 25.** Influence of the distance from blasting centre on the hysteretic energy consumption.

To analyse the influence of the millisecond delay time on the hysteretic energy spectrum, the acceleration signal at a cutting hole charge of 30 kg and a distance from the blasting centre of 48 m is selected. The millisecond delay time is set to 10 ms, 20 ms, 30 ms, and 40 ms. Figure 26 shows that the millisecond delay time has a strong influence on the hysteretic energy spectrum, which is primarily manifested in the following ways. The peak hysteretic energies occur at 20 ms, 30 ms, 10 ms, and 40 ms from small to large. If the destruction caused by plastic cumulative damage to the structure around the blasting centre is considered, the millisecond delay time of 20 ms is the best choice. The natural period of the structure corresponding to the peak hysteretic energy is also different. The natural period of the structure corresponding to the hysteretic energy spectrum with time intervals of 20 ms and 30 ms is 0.05 s, while the natural period with time intervals of 10 ms and 40 ms is 0.06 s. The closer to the natural period of the structure, the more destructive it is.

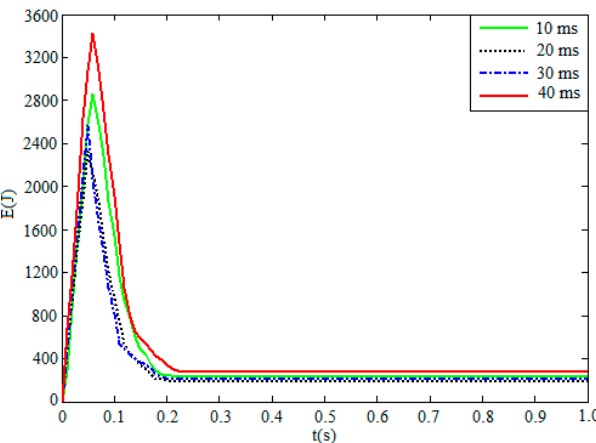

**Figure 26.** Influence of the delay time on the hysteretic energy consumption.

### 5. Evaluation and Control of Blasting Vibration Hazards in Tunnel

There are two primary forms of structural damage under blasting vibration: first passage damage and cumulative damage. According to the above analysis, the hysteretic energy consumption can be used as an index parameter to reflect the cumulative damage, and the instantaneous input energy of the structure can be used for the first passage failure [32–36]. However, the instantaneous input energy cannot accurately describe the cumulative damage, and the hysteretic energy consumption cannot accurately describe the first passage damage [37,38]. Therefore, it is more scientifically rigorous and comprehensive to use these two energies to evaluate vibration hazards. This paper presents an input-hysteresis energy criterion to determine whether the structure is damaged. To verify the validity and rationality of the input-hysteretic energy criterion, 10 sets of data at each position of the tunnel vault, arch waist, and sidewall are used to calculate energy responses, which are shown in Tables 2–4.

**Table 2.** Energy response at the vault.

| No. | Natural Period (s) | Maximum Instantaneous Input Energy (J) | Hysteretic Energy Consumption (J) | Initial Lining State |
|---|---|---|---|---|
| 1 | 0.1428 | 226.99 | 2245.44 | cracked |
| 2 | 0.1462 | 173.57 | 1854.35 | intact |
| 3 | 0.1394 | 127.73 | 1355.32 | intact |
| 4 | 0.1428 | 210.3 | 2678.21 | intact |
| 5 | 0.1378 | 153.84 | 1677.93 | intact |
| 6 | 0.1394 | 154.61 | 1768.86 | intact |
| **7** | **0.1445** | **193.06** | **1459.19** | **intact** |
| 8 | 0.1512 | 150.05 | 1679.37 | intact |
| 9 | 0.1478 | 157.22 | 1712.21 | intact |
| 10 | 0.1378 | 169.17 | 3173.12 | damaged |

Tables 2–4 show that the maximum instantaneous input energy of measuring point No. 7 of the vault is large, but its hysteretic energy consumption is small, as is that of measuring points No. 5 of the arch waist and No. 2 of the sidewall. The measuring points with damage have a common feature: the maximum instantaneous input energy is greater than 200 J, or its hysteretic energy is greater than 3000 J. Therefore, the initial lining of the tunnel causes the first passage damage when the maximum instantaneous input energy exceeds 200 J; once the hysteretic energy consumption exceeds 3000 J, plastic cumulative damage occurs. The input-hysteresis energy criterion explains the phenomenon that the destroyed structure near the blasting centre is easily destroyed, but it is safe according to Chinese safety regulations for blasting vibration. It can evidently supplement and improve

the current safety criterion. However, both the maximum instantaneous energy threshold and the hysteresis energy threshold must be studied in more detail and applied in more engineering applications to verify these results.

**Table 3.** Energy response at the arch waist.

| No. | Natural Period (s) | Maximum Instantaneous Input Energy (J) | Hysteretic Energy Consumption (J) | Initial Lining State |
|-----|-----|-----|-----|-----|
| 1 | 0.1428 | *230.23* | 2319.98 | cracked |
| 2 | 0.1462 | 185.99 | 2039.79 | intact |
| 3 | 0.1394 | 132.4 | 1490.85 | intact |
| 4 | 0.1428 | 145.79 | *3046.03* | damaged |
| **5** | **0.1378** | **192.93** | **1845.72** | **intact** |
| 6 | 0.1394 | 163.82 | 1945.75 | intact |
| 7 | 0.1445 | 138.64 | 1605.11 | intact |
| 8 | 0.1512 | *228.5* | 1847.30 | cracked |
| 9 | 0.1478 | 166.88 | 1883.43 | intact |
| 10 | 0.1378 | 147.84 | *3270.43* | damaged |

**Table 4.** Energy response at the sidewall.

| No. | Natural Period (s) | Maximum Instantaneous Input Energy (J) | Hysteretic Energy Consumption (J) | Initial Lining State |
|-----|-----|-----|-----|-----|
| 1 | 0.1428 | *213.49* | 2800.35 | cracked |
| **2** | **0.1462** | **179.73** | **1891.44** | **intact** |
| 3 | 0.1394 | 130.06 | 1382.43 | intact |
| 4 | 0.1428 | *227.92* | 2731.77 | cracked |
| 5 | 0.1378 | 158.34 | 1711.49 | intact |
| 6 | 0.1394 | 159.18 | 1804.24 | intact |
| 7 | 0.1445 | 135.82 | 1488.37 | intact |
| 8 | 0.1512 | 154.25 | 1712.96 | intact |
| 9 | 0.1478 | 162.02 | 2746.45 | intact |
| 10 | 0.1378 | 148.35 | *3032.58* | damaged |

## 6. Conclusions

This study investigated the arch vibration attenuation law, structural energy response, and safety criterion using blasting vibration monitoring in the Jiaohuayu Tunnel. The primary conclusions of this study are as follows:

(1) The PPV at the vault was always larger than at the arch waist and was greater than at the sidewall regardless of direction (i.e., horizontal tangential, horizontal radial, and vertical). The vertical vibration velocity was the largest; the PPV at the vault decayed faster than the other two; the attenuation of the three-direction PPV at the same measuring point tends to be stable with increasing propagation distance.

(2) The arch waist was where the greatest risk of structural damage occurred at each position of the tunnel based on the dominant frequency. The millisecond delay detonator blasting vibration signal was widely distributed in the frequency domain, but the energy was primarily concentrated at 200 Hz.

(3) When the engineering structure entered the elastic-plastic stage under the action of blasting vibration, the energy dissipation mode began to change, primarily based on damping energy dissipation and hysteretic energy dissipation. This mode is of practical importance to evaluate the safety of cumulative damage of engineering structures and measure the cumulative damage of structures by hysteretic energy.

(4) Existing safety criteria can be supplemented and improved using the maximum instantaneous input energy to measure the first passage damage, the hysteretic energy consumption to measure the cumulative damage, and the input-hysteretic energy criterion to judge the structural failure. The energy threshold of the first passage failure of the test section was 200 J, and the plastic cumulative damage was 3000 J.

The results of this study are important when evaluating the safety of a tunnel's initial lining structure. However, the threshold of maximum instantaneous energy or hysteretic energy consumption will be different for different structures. A lot of experimental data and more engineering practice are required to improve and apply the input-hysteretic energy criterion.

**Author Contributions:** S.H.: writing—original draft, validation, formal analysis; F.X.: conceptualization, methodology, data curation, formal analysis, writing—review and editing; S.T.: formal analysis, writing—review and editing; S.L.: conceptualization, writing—review and editing, funding acquisition. All authors have read and agreed to the published version of the manuscript.

**Funding:** This research was supported by the Fundamental Research Funds for the Central Universities (2017QL05).

**Institutional Review Board Statement:** Not applicable.

**Informed Consent Statement:** Not applicable.

**Data Availability Statement:** Not applicable.

**Conflicts of Interest:** The authors declare that they have no known competing financial interest or personal relationships that could have appeared to influence the work reported in this paper.

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
