# Peer review of "Effects of Blasting Vibrations on an Arch in the Jiaohuayu Tunnel Described by Energy Response Spectrum Analysis"

_applsci, doi:10.3390/app122211395_

Round 1
Reviewer 1 Report
The authors investigated the arch vibration attenuation law, structural energy response and provided safety criteria from their blast vibration studies carried out at one of tunnel.
The manuscript is well written and well presented.
Just I have one suggestion, is there any other good technical term or popularly used to replace the term used like " hance" used in the manuscript.
Reviewer 2 Report
This a well designed study which investigates the effects of blasting vibrations on the tunnel's safety. The manuscript is well organized, and the conclusions are supported by the analysis, and it is worth to be considered for publication in applied science. I would suggest the following to be revised prior publication.
- Introduction: HHT and TEDI -> the authors should define in the manuscript what is the meaning of the acronyms.
- Introduction: The authors provide a detailed literature review. However, the linkage of the research presented in the manuscript with the research gaps is missing. The authors should discuss in the last paragraph the motivation/objectives of the paper and what research gaps is itended to cover.
- 2.1 Blasting Vibration Monitoring Scheme. What was the anchoring agent used by the authors? Do you mean an adhesive agent, i.e. mortar? The authors should explain better that to avoid misunderstandings.
- 2.2 page 3, 3rd line from end: '' Q is the amount of explosive''. Should become explosives.
- page 4: Could the values of K and a be categorized based on the specific geological conditions? For example, could a range for the fitted values be expected for a given terrain and geological condition?
- figure 3: change the legend as xxx direction experimental and xxx direction fitted. Also, for the solid lines the legend xxx direction direction should be corrected, i.e. remove the repeated word direction.
- Eq 3. The authors should add what is the expected error (error order) of the 4-point forward difference method.
Reviewer 3 Report
Please see the file attached

Round 2
Reviewer 2 Report
No further comments.